# Leptin Promotes Expression of EMT-Related Transcription Factors and Invasion in a Src and FAK-Dependent Pathway in MCF10A Mammary Epithelial Cells

**DOI:** 10.3390/cells8101133

**Published:** 2019-09-24

**Authors:** Monserrat Olea-Flores, Miriam Zuñiga-Eulogio, Arvey Tacuba-Saavedra, Magdalena Bueno-Salgado, Andrea Sánchez-Carvajal, Yovani Vargas-Santiago, Miguel A. Mendoza-Catalán, Eduardo Pérez Salazar, Alejandra García-Hernández, Teresita Padilla-Benavides, Napoleón Navarro-Tito

**Affiliations:** 1Laboratorio de Biología Celular del Cáncer, Facultad de Ciencias Químico Biológicas, Universidad Autónoma de Guerrero, Chilpancingo 39090, México; monseolea@uagro.mx (M.O.-F.); miriamzuniga@uagro.mx (M.Z.-E.); arveytacuba@uagro.mx (A.T.-S.); magdans5m@gmail.com (M.B.-S.); andreeeacarbajal15@gmail.com (A.S.-C.); yovanivargas@uagro.mx (Y.V.-S.); 2Laboratorio de Biomedicina Molecular, Facultad de Ciencias Químico Biológicas, Universidad Autónoma de Guerrero, Chilpancingo México; mamendoza@uagro.mx; 3Departamento de Biología Celular, CINVESTAV, Av. Instituto Politécnico Nacional 2508, CDMX 07360, México,; 4Department of Biochemistry and Molecular Pharmacology, University of Massachusetts Medical School, Worcester, MA 01605, USA; Teresita.Padilla@umassmed.edu

**Keywords:** leptin, transcription factors, EMT, invasion, Src, FAK

## Abstract

Leptin is one of the main adipokines secreted in breast tissue. Leptin promotes epithelial–mesenchymal transition (EMT), cell migration and invasion in epithelial breast cells, leading to tumor progression. Although, the molecular mechanisms that underlie these events are not fully understood, the activation of different signaling pathways appears to be essential. In this sense, the effects of leptin on the activation of kinases like Src and FAK, which regulate signaling pathways that activate the EMT program, are not completely described. Therefore, we investigated the involvement of these kinases using an in vitro model for leptin-induced EMT process in the non-tumorigenic MCF10A cell line. To this end, MCF10A cells were stimulated with leptin, and Src and FAK activation was assessed. Specific events occurring during EMT were also evaluated in the presence or absence of the kinases’ chemical inhibitors PP2 and PF-573228. For instance, we tested the expression and subcellular localization of the EMT-related transcription factors Twist and β-catenin, by western blot and immunofluorescence. We also evaluated the secretion and activation of matrix metalloproteases (MMP-2 and MMP-9) by gelatin zymography. Invasiveness properties of leptin-stimulated cells were determined by invadopodia formation assays, and by the Transwell chamber method. Our results showed that leptin promotes EMT through Src and FAK activation, which leads to the secretion and activation of MMP-2 and MMP-9, invadopodia formation and cell invasion in MCF10A cells. In conclusion, our data suggest that leptin promotes an increase in the expression levels of Twist and β-catenin, the secretion of MMP-2, MMP-9, the invadopodia formation and invasion in MCF10A cells in a Src and FAK-dependent manner.

## 1. Introduction

Leptin is a 16 kDa hormone secreted mainly by adipose tissue, and in a lesser extent by the placenta, stomach, fibroblasts, skeletal muscle, normal and tumorigenic mammary epithelial tissue [1]. Leptin plays an important role in the regulation of energy homeostasis, endocrine and immune functions, as well as in lipid and glycolytic metabolism [2,3]. The overexpression of leptin and its receptor, the ObR, has been associated to tumor progression-related events in breast cancer [4,5,6]. Leptin exerts its effects through binding to the ObR receptor, promoting the activation of several signaling pathways such as MAPK, PI3K/AKT and JAK/STAT [3,7,8]. The signaling pathways activated by leptin regulate cell proliferation, migration, invasion and the epithelial–mesenchymal transition (EMT) [4,5,9,10,11].

The EMT is a reversible cellular trans-differentiation program from epithelial to a mesenchymal phenotype, which provides greater plasticity and dynamism to the cells [3,12,13,14]. EMT is a physiological process necessary for embryonic development, wound healing and tissue regeneration; however, it is also associated with cancer progression [12,13]. At the molecular level, the EMT is characterized by the loss of epithelial markers such as E-cadherin, ZO-1 and claudins, and the acquisition of mesenchymal markers including N-cadherin, vimentin, Hic-5 and matrix metalloproteases (MMPs) [3,15]. To initiate the EMT program, signals are required to activate specific transcription factors (TFs) such as Snail, Slug, Zeb and Twist [13]. These TFs directly regulate genes involved in cell adhesion, polarity and cytoskeleton reorganization [16]. Overexpression of these TFs in epithelial cells induces either partial or complete changes from an epithelial to mesenchymal morphology, such as increased motility, invasive capacity and resistance to chemotherapy [17,18,19]. In normal conditions, the epithelial cells maintain adherent junctions, where E-cadherin forms a complex with β-catenin in the cell membrane [3,20]. During the EMT β-catenin translocates from the membrane to the nucleus where it establishes a complex with the TFs TCF/LEF. This complex induces the expression of EMT-related genes, by recognizing the consensus sequence T/A-CAAAG found in the HMG box in the promoter region of genes such as *MYC, CCND1, SNAI1* and *TWIST1* [21,22,23]. Among these, Twist is a 28 kDa, a basic helix-loop-helix (bHLH) domain-containing TF, necessary for cell invasion and cancer progression [24]. The known mechanism by which Twist favors cell migration and invasive capabilities of the cells is through the binding to the consensus sequence CANNTG of the *CDH1* promoter to repress the expression of E-cadherin, leading to the loss of intercellular adhesions [25]. Twist is also necessary for the formation of invadopodia [26].

Cell invasion is a process associated to EMT, which requires the degradation of the extracellular matrix (ECM) by the tumor cell to allow its infiltration to the adjacent tissue [27]. To achieve this, one of the primary structures formed during the local invasion is the invadopodia, which are membrane protrusions rich in actin puncta [28,29]. Invadopodia formation is required for the local degradation of ECM components through the secretion and activation of MMP-2, MMP-9, and MMP-14 [28,29,30]. MMP-2 and MMP-9 (also called gelatinase A and B, respectively) have been correlated with the invasive stage of carcinomas due to their ability to degrade gelatin and collagen type IV, the main components in the basal membrane [29,31]. Experimental evidence suggests that MMP-2 and MMP-9 also contribute to the initiation and progression of breast cancer by cleaving and activating diverse proteins involved in angiogenesis, invasion and metastasis [32]. In addition, a high activity of MMP-2 and MMP-9 has been observed in the serum and tissues of patients with breast cancer compared to healthy individuals [33,34]. Several signaling pathways, such as WNT, PI3K/AKT, MAPK, JNK, focal adhesion kinase (FAK) and Src, are activated during the remodeling of the ECM [35]. Among these kinases, the secretion and activation of MMP-2 and MMP-9 in breast cancer cell lines is dependent on the cytosolic tyrosine kinases Src [36,37,38,39], and FAK [40,41]. Both enzymes are necessary for proliferation, cell migration, invasion and metastasis, at least in part for their participation in the regulation of the expression and activation of MMPs [36,40,41,42].

Previous reports using the non-tumorigenic breast epithelial cell line MCF10A demonstrated that leptin induces a partial EMT, where the cells change from an epithelial to a mesenchymal morphology [43]. This phenotype was characterized by a leptin-induced activation of FAK and ERK, which correlated with an increase in the expression of the intermediate filament vimentin, as well as the relocation of E-cadherin from membrane to cytoplasm [43]. These events favored the collective cell migration of MCF10A cells [43]. However, the mechanisms by which leptin activate other signaling pathways, and how the kinases contribute to EMT are not fully understood. Therefore, the aim of this research was to determine the role of Src and FAK in the expression of EMT-related transcription factors and invasion in MCF10A cells stimulated with leptin. We found that leptin activates Src and FAK, leading to a variety of EMT-associated events, such as expression of EMT-related the TFs, Twist and β-catenin, as well as MMPs secretion, invadopodia formation and invasion in the non-tumorigenic MCF10A epithelial cell line. This report provides insights on the signaling pathways operating downstream of the leptin-induced EMT, and their ultimate functional effectors, which contributes to this process.

## 2. Materials and Methods

### 2.1. Materials

Recombinant human leptin, FAK (PF-573228) and Src (PP2) inhibitors were obtained from Sigma-Aldrich (St Louis, MO, USA). Mouse anti-actin, rabbit anti-FAK and anti-Src antibodies were purchased from Santa Cruz Biotechnology (Santa Cruz, CA, USA). Rabbit anti-GAPDH antibody was from Cell Signaling Technologies (Danvers, MA, USA). The phospho-specific antibody against FAK, rabbit anti-pY397, was obtained from Invitrogen (Carlsbad, CA, USA). The phospho-specific antibody against Src, rabbit anti-pY418, was obtained from MyBiosource (San Diego, CA, USA). The secondary HRP-conjugated antibodies were from Millipore (Billerica, MA, USA), and the anti-mouse and anti-rabbit conjugated with Alexa Fluor 488 antibodies were from Invitrogen (Carlsbad, CA, USA). TRITC conjugated phalloidin was purchased from Cytoskeleton (Denver, CO, USA).

### 2.2. Cell Culture

The non-tumorigenic mammary epithelial cell line MCF10A (ATCC, Manassas, VA, USA) was cultured in DMEM/F12 media (50:50, V:V; Sigma-Aldrich, St Louis, MO, USA) supplemented with 5% fetal bovine serum (FBS) and 1% antibiotics (penicillin G/Streptomycin, Gibco, Waltham, MA, USA) in a humidified atmosphere containing 5% CO_2_ at 37 °C. For experimental purposes, cell cultures were serum-starved for 4 h before treatment with FAK or Src inhibitors and/or leptin. Cell cultures were used between passages 3–15.

### 2.3. Cell Stimulation

MCF10A cell cultures were washed with PBS, and then treated with FAK (10 µM) or Src (10 µM) inhibitors and/or leptin for the times and concentrations indicated in the figure legends. Cell cultures were grown to confluence in 60 mm plates containing 3 mL of DMEM/F12 for each experimental condition. Cell stimulation was terminated by removing the medium, and solubilizing the cells in 0.5 mL of ice-cold radioimmune precipitation assay (RIPA) buffer, containing 50 mM HEPES pH 7.4, 150 mM NaCl, 1 mM EGTA, 1 mM sodium orthovanadate, 100 mM NaF, 10 mM sodium pyrophosphate, 10% glycerol, 1% Triton X-100, 1% sodium deoxycholate, 1.5 mM MgCl_2_, 0.1% SDS and 1 mM phenylmethylsulfonyl fluoride (PMSF).

### 2.4. Western Blot

Whole cell lysates (20 µg) were resolved on 10% SDS-polyacrylamide gels and transferred to nitrocellulose membranes. To block non-specific binding, the membranes were incubated in 5% (*w*/*v*) non-fat dried milk in TBS/T (0.05% Tween 20 in TBS) for 2 h at room temperature (RT). Membranes were then incubated with the primary antibodies, anti-Twist, anti-β-catenin, anti-actin, anti-pY397, anti-FAK, anti-pY418 and anti-Src overnight at 4 °C (1:1000 dilution). Next, the membranes were washed with TBS/T and incubated with HRP-conjugated antibodies (1:5000 dilution) for 2 h, at RT. Membranes were developed using an enhanced chemiluminescence detection system from Bio-Rad (Hercules, CA, USA).

### 2.5. Immunofluorescence and F-Actin Staining

MCF10A cells were seeded on glass coverslips, grown to 70% confluence, and stimulated either with or without leptin for 3, 6, 12, 24 and 48 h. Cells were fixed for 5 min with 4% paraformaldehyde in PBS and permeabilized with 0.2% Triton-X 100 in PBS at RT. For immunofluorescence (IF) assays, non-specific staining was blocked with 3% bovine serum albumin in PBS for 1 h at RT. Then, the cells were incubated 2 h at RT with the anti-Twist, anti-β-catenin antibodies (1:250 dilution), followed by a 2 h incubation at RT with an anti-rabbit conjugated to Alexa Fluor 488 secondary antibody (1:400 dilution). For F-actin staining, cells were incubated with TRITC-conjugated phalloidin (1:1000 dilution) for 30 min at RT. Nuclei were counterstained with 4’,6-diamidino-2’-phenylindole and samples were mounted with Fluoroshield/DAPI media (Sigma-Aldrich), and imaged with an Olympus BX43 microscope, using the 100× immersion objective. Images were analyzed with ImageJ software, version 1.44p (NIH, Bethesda, MD, USA) [44].

### 2.6. Gelatin Zymograms

To assess MMP-2 and MMP-9 activity, MCF10A cells were stimulated with leptin and/or kinases inhibitors in serum-free DMEM/F12 medium for the time and concentrations indicated in the figure legends. The conditioned medium was collected and concentrated using 30 kDa cutoff ultra-centrifugal filter units (Amicon, Merck-Millipore, Burlington, MA, USA). Protein concentration was determined by the Bradford method [45], and 20 µg of concentrated supernatant of each condition were assayed for proteolytic activity on gelatin-substrate gels [46]. Briefly, samples were mixed with non-reducing buffer containing 2.5% SDS, 1% sucrose and 4 mg/mL phenol red, and separated in 8% acrylamide gels co-polymerized with 1 mg/mL gelatin, as previously described [20]. After electrophoresis at 72 V for 2.5 h, the gels were rinsed twice in 2.5% Triton X-100, and then incubated in 50 mM Tris-HCl pH 7.4 and 5 mM CaCl_2_ assay buffer at 37 °C for 24 h. Gels were fixed and stained with 0.25% Coomassie Brilliant Blue G-250 in 10% acetic acid and 30% methanol. Proteolytic activity was detected as clear bands against the background stain of undigested substrate in the gel. Quantification was performed using ImageJ software, version 1.44p (NIH, Bethesda, MD, USA) [44].

### 2.7. Invadopodia Formation Assays

MCF10A cells were treated with leptin 400 ng/mL and then seeded in coverslips that were previously treated for invadopodia formation assays, as described [47]. Briefly, the coverslips were incubated with 20% sulfuric acid overnight and sterilized with 96% ethanol. Then, the coverslips were covered with 50 μg/mL of poly-L-Lysine diluted in PBS, followed by a treatment with 0.5% glutaraldehyde in PBS, and a final incubation with 0.2% gelatin solution in PBS at 37 °C for 30 min. Pre-treated MCF10A cells were cultured for 6 h in serum supplemented medium. The cells grown on the coverslips were fixed and permeabilized with 4% formaldehyde/0.5% Triton X-100 in PBS, blocked with 3% BSA and incubated with phalloidin-TRITC for 30 min at 37 °C. Finally, the coverslips were mounted using Fluoroshield/DAPI media and visualized in a Olympus BX43 microscope (Olympus Corporation; Allentown, PA, USA), using the 100× immersion objective. Images were analyzed with Image J software, version 1.44p (NIH).

### 2.8. Cell Invasion Assays

Matrigel invasion assays were performed following the Transwell chamber method [48], using 24 well plates containing inserts of 8 μm pore size (Corning, Kennebunk, ME, USA). Briefly, 30 µL of Matrigel (Corning) was added into the inserts and kept at 37 °C for 30 min to form a semisolid matrix. MCF10A cells were treated for 2 h with 10 μM Cytosine β-D-Arabinofuranoside (AraC) to inhibit cell proliferation during the experiment. Then, cells were treated or not with the 10 µM of Src inhibitor and plated at 1 × 10^5^ cells per insert in serum-free medium on the top chamber. The lower chamber of the Transwell contained 600 µL DMEM supplemented or not with leptin 400 ng/mL. Cells were incubated for 48 h at 37 °C in a 5% CO_2_ atmosphere. Following incubation, cells and Matrigel on the upper surface of the Transwell membrane were gently removed with cotton swabs. Invading cells on the lower surface of the membrane were washed and fixed with methanol for 5 min, and stained with 0.1% crystal violet diluted in PBS. Cell quantification was performed using a hemocytometer, and an Olympus BX43 microscope with the 100× objective.

### 2.9. Statistical Analysis

Results are expressed as the mean ± SD. Data was statistically analyzed using a one-way ANOVA and the comparisons were performed using Newman–Keuls and Dunnett´s multiple comparison tests. A statistical probability of *p* < 0.05 was considered significant.

## 3. Results

### 3.1. Leptin Induces FAK and Src Activation in MCF10A Non-Tumorigenic Epithelial Breast Cells

Kinase activation is a fundamental process for cancer progression and metastasis [49,50]. Src and FAK activation is triggered by the autophosphorylation at Y418 and Y397, respectively [49,51,52]. Emerging evidence from our lab and others showed that these kinases might also contribute to early events during EMT [53,54,55,56]. We reported previously that leptin induces the activation of FAK, and in consequence a partial EMT was detected in MCF10A cells [43]. In this work we further explored whether leptin induces the activation of FAK and Src in MCF10A cells, and how these kinases were associated to the expression of the downstream EMT-related TFs, Twist and β-catenin. Therefore, we initiated our studies by analyzing the activation of Src and FAK, as both kinases are associated to the induction of Twist and β-catenin [57,58]. To this end, MCF10A cultures were stimulated with leptin 400 ng/mL and protein extracts were obtained at different time points for western blot analyses of Src and FAK activation. Figure 1 shows that leptin induces Src phosphorylation in Y418 between 15–30 min of stimulation (Figure 1A,C), while FAK Y397 phosphorylation occurs within 10–15 min (Figure 1B,D), consistent with our previous report [43]. These data suggest that FAK and Src, two kinases involved in cell proliferation and migration are activated by leptin. To further clarify the role of Src in the activation of FAK, the Src inhibitor PP2 was used and FAK phosphorylation was analyzed by western blot. We observed that the activation of FAK depends on the kinase activity of Src (Figure 1E,G). Interestingly, a feedback loop seems to modulate the activity of these kinases, since inhibition of FAK with PF573228 also led to a decrease in Src phosphorylation, suggesting that FAK activity is also essential in the activation of Src (Figure 1F,H). These data, suggest a positive feedback mechanism during the activation of both kinases in MCF10A cells stimulated with leptin.

### 3.2. Leptin Regulates the Expression and Subcellular Localization of Twist and β-Catenin in MCF10A Epithelial Breast Cells

Twist and β-catenin are TFs important players during EMT, tumor progression, chemoresistance and cancer stem cell (CSC) maintenance [59,60,61]. Importantly, leptin induces the expression of Twist and β-catenin in breast cancer cells [9,10]. Therefore, we evaluated whether leptin induces the expression of these two TFs in a cellular model of non-tumor mammary epithelium. To determine the expression and subcellular localization of Twist and β-catenin in MCF10A cells stimulated with leptin, western blot and immunofluorescence analyses were performed. An early maximum peak of Twist expression was detected after 3 h of leptin treatment, which gradually decreased at 12 h and was lost after 48 h of the stimulus (Figure 2A). Quantification and statistical analyses of the western blots (Figure 2B), as well as epifluorescence microscopy imaging confirmed these results, and demonstrated that basal levels of Twist is located in the nucleus at 0 h, however, higher levels of twist at 3 and 6 h of leptin treatment were observed (Figure 2D, white arrows). Conversely, β-catenin expression in MCF10A cells stimulated with leptin remained stable during the first 24 h of treatment (Figure 2A,C), but was induced after 48 h of stimulus, as corroborated by statistical analyses (Figure 2C). IF analyses showed that β-catenin was located at the cell membrane in non-treated cells (Figure 1E, red arrows); within 3 h of treatment, β-catenin was predominantly located in the nucleus (Figure 1E, white arrows). After 6 h of leptin treatment, β-catenin seemed to acquire an equilibrium state between the cell membrane and the nucleus, being predominantly located at the membrane; this equilibrium was lost after 48 h when β-catenin returns to the nucleus (Figure 2E). This data strongly suggest that leptin induce an early activation and nuclear translocation of Twist and β-catenin, and may act as TFs of EMT-associated events.

### 3.3. Leptin Regulates the Expression and Subcellular Localization of Twist and β-Catenin in a Src and FAK-Dependent Manner in MCF10A Cells

Twist and β-catenin are downstream effectors of different kinases and signaling pathways [24]. We focused on the effect of Src and FAK in the activation of these TFs, as both kinases are associated to EMT-related events [15,43,51]. To address this, we took advantage of the specific Src and FAK inhibitors, PP2 or PF-573228, respectively. MCF10A cells were pre-treated with either inhibitor, and then stimulated with leptin for 3 and 48 h. Western blot and IF analyses showed that in the presence of the Src inhibitor (PP2) the leptin-dependent increase observed for the two TFs, Twist and β-catenin, was abolished. Control blots showing the loss of pY416-Src and pY397-FAK in the presence of inhibitors, PP2 and PF573228 are shown (Figure 3A–C,G). These data suggest that leptin may activate these TFs, at least partially, via Src signaling pathway. On the other hand, the inhibitor for FAK (PF-573228) only had a preventive effect on the leptin-induced expression of Twist, as this TF was expressed at similar levels observed in non-treated control cells (Figure 3D,E,H). No changes in the levels of expression of β-catenin induced by leptin were observed upon FAK inhibition (Figure 3D,F). Moreover, the leptin-induced nuclear distribution of β-catenin was maintained in the presence of the FAK inhibitor (Figure 3H). These data suggest that FAK signaling might be involved in the expression and transcriptional activation of Twist, but not in the β-catenin pathway.

### 3.4. Leptin Induces MMP-2 and MMP-9 Secretion in a Src- and FAK-Dependent Fashion in MCF10A Cells

MMPs are a family of endopeptidases capable to degrade the ECM components, thus they are fundamental players for tumor progression, by promoting invasion through the basement membrane and interstitial matrix, angiogenesis and growth of tumor cells [62,63]. MMP-2 and MMP-9 expression is elevated in malignant tumors and largely contribute to the ability of tumor cells to metastasize, as they degrade collagen type IV, which is the main component of the basement membrane [64]. To date, little information is available about the regulation of MMP-2 and MMP-9 expression and secretion during EMT [65], and less is our knowledge on how leptin mediates this process. Therefore, to determine whether leptin induces the MMP-2 and MMP-9 secretion and activation, gelatin zymograms were performed. MCF10A cells were treated with increasing concentrations of leptin for 24 h (Figure 4A), the supernatant was collected, concentrated and 20 µg of protein were used to assay the activity of both MMPs. Densitometric and statistical analyses, demonstrated that 400 ng/mL of leptin induced the highest level of activation of both MMPs (Figure 4B,C). Therefore, we chose this concentration for further experimentation, and treated the cells for different times (Figure 4D–F). In this instance, we detected that activity of both MMPs further increased after 6 h of leptin treatment (Figure 4D). It is noteworthy that MMP-2 activation was greater than MMP-9. Then, to establish whether the Src and FAK regulate the secretion of MMP-2 and MMP-9, the inhibitors PP2 and PF-573228 were used. Zymograms showed that inhibition of both kinases impaired the leptin-induced activation of MMP-2 and MMP-9; densitometric analyses of three independent biological replicates confirmed that the inhibitory effect was statistically significant (Figure 4G–L). These data suggest that the leptin-induced activation of MMP-2 and MMP-9 relies in the Src and FAK signaling pathways.

### 3.5. Leptin Induces the Formation of Stress Fibers in a Src- and FAK-Dependent Manner in MCF10A Cells

Stress fibers are essential for cell motility and the maintenance of structure [66]. During EMT the remodeling of thin cortical actin filament bundles to thick, parallel, contractile bundles that disassembled slowly is necessary to allow the progressive changes in cell morphology [67]. Our group and others have shown that during leptin-induced EMT of MCF10A cells [43], and in several cancer cell lines [51,68,69], this hormone promotes the formation of stress fibers [70,71]. However, understanding of how this process is regulated is still limited. Thus, we asked whether the leptin-induced activation of the kinases Src and FAK had an effect in the formation of stress fibers in MCF10A cells. First, we verified that leptin indeed promoted the formation of stress fibers. As expected the leptin treated cells presented more and thicker actin filaments with a linear distribution, compared to the less and circular fibers observed in non-treated control cells (Figure 5A,B [43]). This leptin induced-effect was abolished in MCF10A cells that were pre-treated with the Src (PP2, Figure 5A,C), but not with the FAK (PF-573228) inhibitor (Figure 5B,C). We showed previously that leptin treatment induces a morphological change from a cuboid epithelial morphology to an enlarged mesenchymal phenotype in MCF10A cells [43]. We observed that treatment with inhibitors for both kinases prevented the morphological changes observed with leptin alone (Figure 5A–C). These data suggests that modifications in the actin cytoskeleton during EMT may rely in the activation of these two kinases by two independent mechanisms.

### 3.6. Leptin Induces Invadopodia Formation in a FAK and Src-Dependent Pathway in MCF10A Cells

Invasive properties of the cells correlate with a dynamic membrane structure rich in actin [72]. These membrane protrusions, called invadopodia, function as matrix adhesion sites [73] able to degrade the ECM components, and are characteristic of invasive tumor cells [74]. An actin-core is formed during the initial steps of invadopodia assembly, which acquires a perpendicular distribution to the ECM and the membrane and can be frequently observed as actin puncta [75]. Considering the effect that leptin excreted in the actin cytoskeleton (Figure 5A–C), we sought to determine whether leptin is also capable to induce the invadopodia formation in MCF10A cells. Therefore we performed in vitro invadopodia formation assays, using Alexa Fluor 488-labeled gelatin assays, which allow us to detect both, the presence of invadopodia and their activity using fluorescence microscopy [76]. Our results show that control cells did not present actin puncta, and were unable to degrade the gelatin matrix (Figure 5D). However, upon leptin treatment, the cells presented actin puncta formation, which colocalized with areas of gelatin degradation (observed as black dots, indicated by white arrows; Figure 5D).

Fluorescence microscopy analyses revealed that Src had an effect in the formation of thick stress fibers, whereas FAK appeared to be not essential in this process (Figure 5). However, both kinases are likely relevant for the leptin-induced morphological changes observed in MCF10A cells. Thus, we hypothesized that both kinases may also be involved in invadopodia formation, as this structure contributes largely to the morphological and migratory phenotypes that emerge during EMT [77,78]. Therefore, to determine the role of Src and FAK in the invadopodia formation in MCF10A cells stimulated with leptin, we performed similar gelatin degradation experiments in the presence and absence of the kinases’ inhibitors. Our results show the actin puncta formation and an increase in the degradation of gelatin, an indicative of invadopodia formation, in the leptin-treated cells (Figure 6). When the cells were treated with either the Src- or the FAK-specific inhibitors, actin puncta formation and gelatin degradation was not observed (Figure 6). It is noteworthy that MMP-14 is a predominant transmembrane protease activated in the regions where invadopodia is formed [79]; thus we cannot overrule the possibility that both kinases also promotes the activation of MMP-14 in a leptin-dependent manner.

### 3.7. Leptin Induces Cell Invasion in a Src-Dependent Pathway in MCF10A Cells

At this point, we have evidence of the requirement of Src and FAK in the secretion and/or activation of at least three MMPs relevant for EMT. Previously, our group showed that FAK is necessary for the acquisition of enhanced migratory capabilities driven by leptin in MCF10A cells [43]. Therefore, here we evaluated the contribution of Src in cell invasion in MCF10A cells stimulated with leptin. To prevent cell proliferation, the cells were pre-treated with AraC and invasion assays were performed in Transwell chambers [48], with MCF10A cells that were treated or not with leptin and PP2. Figure 7 shows that control cells had reduced mobility, and were not capable of colonizing the matrix in the Transwell, while leptin-treated cells were capable of invading this area. As expected, Src appeared to be an essential kinase in this process, as the leptin-induced invasion properties of MCF10A cells treated with PP2 was ablated. These data suggest that Src is a relevant kinase for the establishment of the migratory and invasive phenotype during early stages of EMT.

## 4. Discussion

EMT is a reversible cellular process by which epithelial cells undergo trans-differentiation events from an epithelial to a mesenchymal phenotype, where EMT-related TFs are key molecules [3]. EMT is also associated with tumor progression and the invasive and metastatic phenotype in cancer cells [14]. In this work, we investigated the role of leptin during EMT, as it is a hormone primarily secreted by adipocytes, which is secreted in a lesser extent, in normal and tumorigenic mammary tissue [11,80]. Studies have established the association between the overexpression of leptin and its receptor, ObR, during breast cancer progression [4,5,6]. However, there are still gaps in our understanding on the mechanisms by which leptin contributes to EMT and during the onset of breast cancer. Expression and hyperactivation of various molecules (kinases, TFs, MMPs and adhesion proteins) are associated with the activation of the EMT program; a couple of examples are the kinases Src and FAK [81]. In this study, we evaluated the effect of leptin in the activation of these kinases, and how these signaling molecules regulate the expression of EMT-related TFs, and additional invasion-related processes in the non-tumorigenic mammary epithelial cell line MCF10A. Src and FAK activation was evaluated through the phosphorylation levels of Y418 and Y397, respectively, in MCF10A cells stimulated with leptin 400 ng/mL. Our results showed early maximal activation peaks for both kinases within 10–15 min of leptin stimulation. These data was consistent with our previous report, where leptin induced the activation of FAK between 10–30 min of stimulation in MCF10A, MCF7 and MDA-MB-231 cells [43,51]. Importantly, by using Src and FAK specific inhibitors, we found a feedback mechanism, which may further modulate downstream signals. Therefore, the mechanism by which leptin activates up- and down-stream effector molecules of Src and FAK in MCF10A cells should be explored. Various alternatives arise from previous studies. For instance, in circulating angiogenic cells, leptin induces Src activation through the αvβ5 integrins pathway [82]. Likewise, in the colon adenocarcinoma SW480 cell line, FAK activation induced by leptin is also dependent on integrins activation [83]. In contrast, in MDA-MB-231 cells Src activation relies on SHP2 to bind to the phosphorylated Y985 of the ObR receptor [84]. Therefore, further studies should be directed to elucidate the mechanism of activation of both kinases in response to leptin in the MCF10A epithelial cell line.

The EMT program activation is regulated by specific TFs such as Twist and β-catenin, which are associated with cell invasion and metastasis [13,14]. In this study, we showed that leptin-induced Twist expression is an early event that may contribute to the initial establishment of EMT. These results are in agreement with studies performed in the breast cancer cell lines MCF7 and SK-BR-3, where leptin also induced the expression of this TF [9,10]. It is noteworthy that the slight differences in the activation kinetics for these TFs between cancer cell lines and the non-tumorigenic cells used here may be due to the altered cellular background of transformed cell lines. Furthermore, the fact that MCF10a cells undergo EMT-associated events upon leptin stimulation might be a good reflection of the early events that occurs during cancer and metastasis progression. Here, we demonstrated that Twist expression and localization is regulated by Src and FAK. These novel activation pathways for Twist contribute to our understanding of the signaling and transcriptional events associated to this factor during leptin-induced EMT. For instance, in MDA-MB-231 cells Src activates STAT3, a TF that regulates *TWIST1* expression [58]. Leptin also activates STAT3 in a JAK2-dependent manner, and allows its dimerization and nuclear translocation where it regulates the expression of the *TWIST1* gene [85,86]. In PCa cells, which are of prostate origin but derived from their metastatic site, the bone, the inhibition of FAK decreased the activation of the Hypoxia-inducible factor 1-α (HIF-1α), which also regulates *TWIST1* [87]. On the other hand, Src and FAK also regulate the nuclear translocation of β-catenin, whose primary function is to maintain adherent junctions by interacting with E-cadherin, α-catenin and the actin cytoskeleton. Once β-catenin is translocated into the nucleus, it acts as TF forming complexes with DNA binding proteins such as TCF/LEF [88,89]. In this study, we observed that β-catenin expression increased, and was located in the nucleus of leptin-stimulated cells for 3 h. These findings may correlate to the decrease in the expression or changes in the location of proteins from the adherent junctions, in particular, E-cadherin and β-catenin [90]. The leptin-induced overexpression of both TFs and their nuclear localization could be related to the expression of EMT marker genes, such as MMPs expression, which are essential for enhanced invasive capabilities of mammary epithelial cells MCF10A, as previously described [91,92,93].

Actin re-organization and the formation of stress fibers are essential events for cell migration, and are considered the hallmark of EMT [94,95]. Studies have shown that leptin induces remodeling of the cytoskeleton and morphological changes in mammary epithelial cells [43,71]. Further, in MDA-MB-231 breast cancer cells FAK and Src also regulate the formation of stress fibers [96]. We demonstrated that leptin induces the formation of stress fibers in an Src-dependent manner, and that both kinases contributed to the morphological changes observed upon leptin treatment. One of the functional consequences of EMT is invasion, which initiates with invadopodia formation to promote the local degradation of ECM, through the recruitment of MMPs, such as MMP-2 and MMP-9, and MMP-14 [75,97,98]. Here, we observed that leptin induces the expression, secretion and activation of these MMPs, and promotes the formation of invadopodia in an Src- and FAK-dependent manner. In cellular models of breast cancer, both MMP-2 and MMP-9, participate in invasion, metastasis and angiogenesis [32]. A study conducted in multiple tumor cell lines showed that the expression of MMP-9 is regulated by the FAK-ERK1/2-Akt pathway, through the regulation of p65 and c-Fos [99]. While in SW480 cells the expression of MMP-2 is dependent on the FAK/Src and ERK1/2 pathways [100]. Similarly, a study conducted in the A549 human non-small-cell lung carcinoma cell line, FAK-Src, ERK1/2 and β-catenin regulate the expression of MMP-2 and MMP-9 [101]. Interestingly, these EMT markers could also be regulated by additional signaling pathways involving GSK3β, Akt or PKM2 [9,10]. However, additional studies are required to clarify this hypothesis.

## 5. Conclusions

Our results demonstrate that leptin activates Src and FAK, which in turn regulate the expression of the EMT-related TFs β-catenin and Twist. Morphological remodeling and invadopodia formation, as well as MMPs secretion and activation, appear to be also effectors of leptin-induced EMT. All together, these EMT markers play a key role during the cell invasion of tumor cells (Figure 8).

## Figures and Tables

**Figure 1 cells-08-01133-f001:**
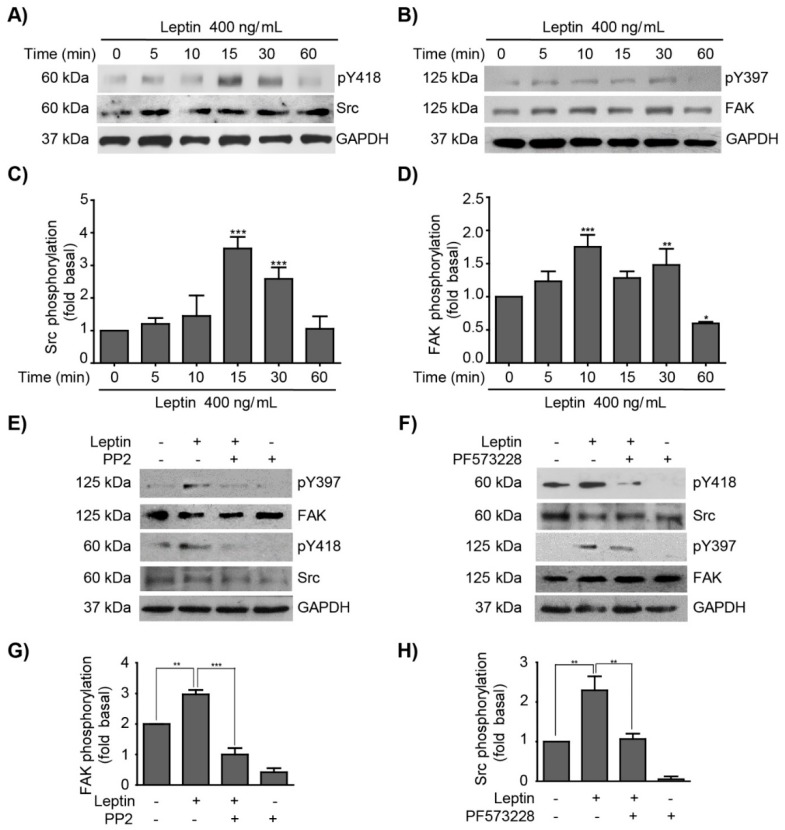
Leptin activates Src and FAK in the non-tumorigenic breast epithelial cell line MCF10A. Representative Western blots of whole cell extracts obtained at different time points from MCF10A cells stimulated with 400 ng/mL of leptin. (**A**) Effect of leptin in Src activation was detected with an anti-p-Y418 antibody and was compared to total Src. (**B**) Effect of leptin in FAK activation was detected with an anti-p-Y397 antibody and was compared to total FAK. GAPDH was used as loading control. Densitometric analyses of (**C**) Src, and (**D**) FAK phosphorylation dependent on leptin. MCF10A cells were pre-treated with Src (PP2) or FAK (PF-573228) inhibitors, and subsequently with leptin 400 ng/mL. Representative western blots of Src phosphorylated in Y418 (**E**), and FAK phosphorylated in Y397 (**F**). Control blots showing loss of pY418-Src and pY397-FAK in the presence of inhibitors, PP2 and PF573228 are shown. GAPDH was used as a loading control. Densitometric analyses of leptin-dependent (**G**) FAK, and (**H**) Src phosphorylation in the presence of the inhibitors. The values are shown in means ± SD of three independent experiments and are expressed as changes with respect to the control (unstimulated cells). The asterisks indicate the comparison made with respect to the control. * *p* < 0.05, ** *p* < 0.01 and *** *p* < 0.001 by one-way ANOVA (Dunnett’s and Newman-Keuls’s test).

**Figure 2 cells-08-01133-f002:**
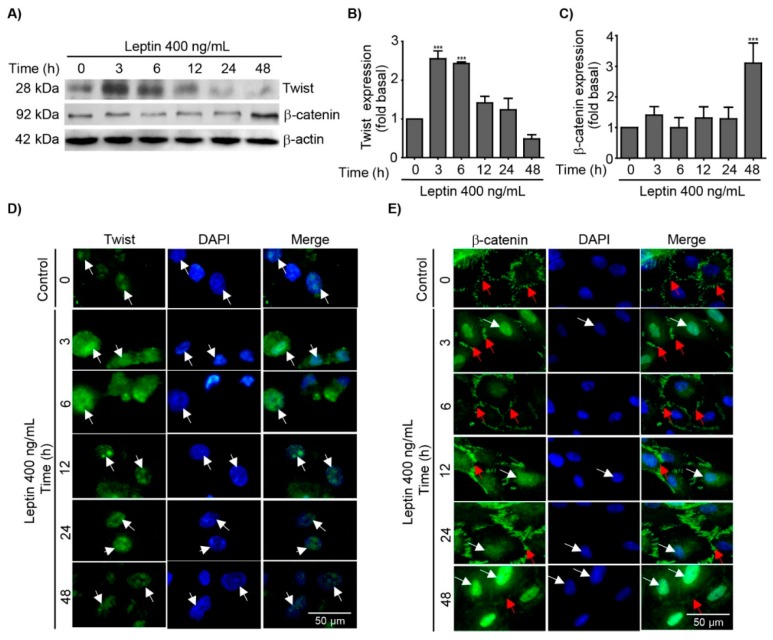
Leptin induces the expression of the epithelial–mesenchymal transition (EMT)-related transcription factors, Twist and β-catenin in MCF10A cells. MCF10A cells were treated with leptin 400 ng/mL for different times and the expression and localization of Twist and β-catenin was analyzed. (**A**) Representative western blot of Twist and β-catenin. β-actin was used as a loading control. Densitometric and statistical analysis of the bands obtained by western blot for Twist (**B**) and β-catenin (**C**); the values are shown in means ± SD of three independent experiments, and are expressed as changes in respect to the control (unstimulated cells). The asterisks indicate the comparison made with respect to the control. *** *p* < 0.001 by one-way ANOVA (Dunnett´s test). Representative immunofluorescence microscopy images showing in green, the expression of Twist (**D**) and β-catenin (**E**), the nucleus was counterstained with DAPI. White arrows indicate the nucleus; red arrows indicate cell membranes. Images were acquired using the 100× magnification.

**Figure 3 cells-08-01133-f003:**
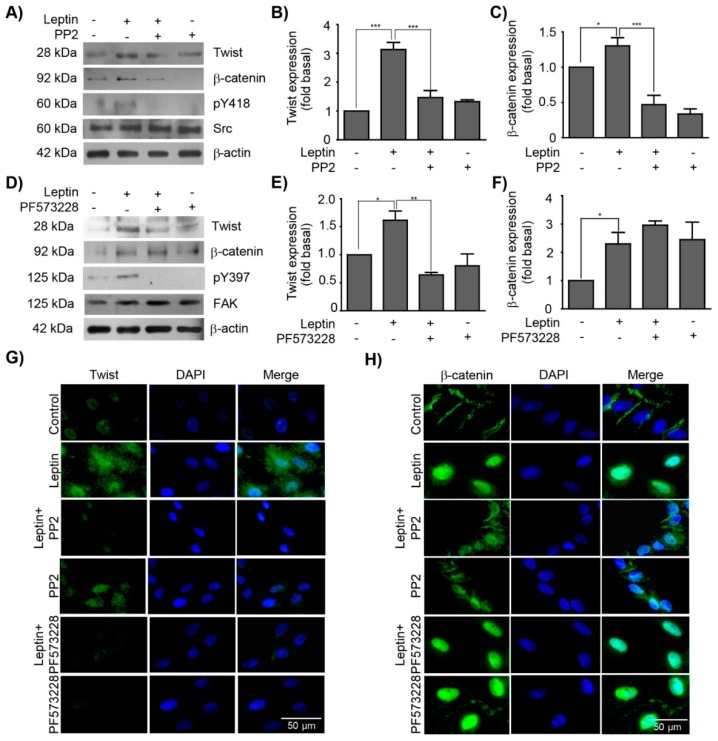
Src and FAK regulate the expression and subcellular localization of Twist and β-catenin. MCF10A cells were pre-treated with Src (PP2) or FAK (PF-573228) inhibitors, and subsequently with 400 ng/mL of leptin. (**A**) Representative western blots of Twist and β-catenin expression upon Src inhibition. Densitometric analyses of (**B**) Twist and (**C**) β-catenin expression upon Src inhibition. (**D**) Representative western blots of Twist and β-catenin expression upon FAK inhibition. Densitometric analyses of (**E**) Twist and (**F**) β-catenin expression upon FAK inhibition. β-actin was used as a loading control. The values represent the mean ± SD of three independent experiments and are expressed as changes with respect to the control (unstimulated cells). The asterisks indicate the comparison made with respect to the control. * *p* < 0.05 and *** *p* < 0.001 by one-way ANOVA (Newman–Keuls’s test). Representative images of epifluorescence microscopy using a 40× magnification. MCF10A cells were pre-treated with PP2 and PF-573228 and subsequently stimulated with leptin (400 ng/mL). The subcellular localization of (**G**) Twist and (**H**) β-catenin is shown in green, and in blue the DNA was counterstained with DAPI.

**Figure 4 cells-08-01133-f004:**
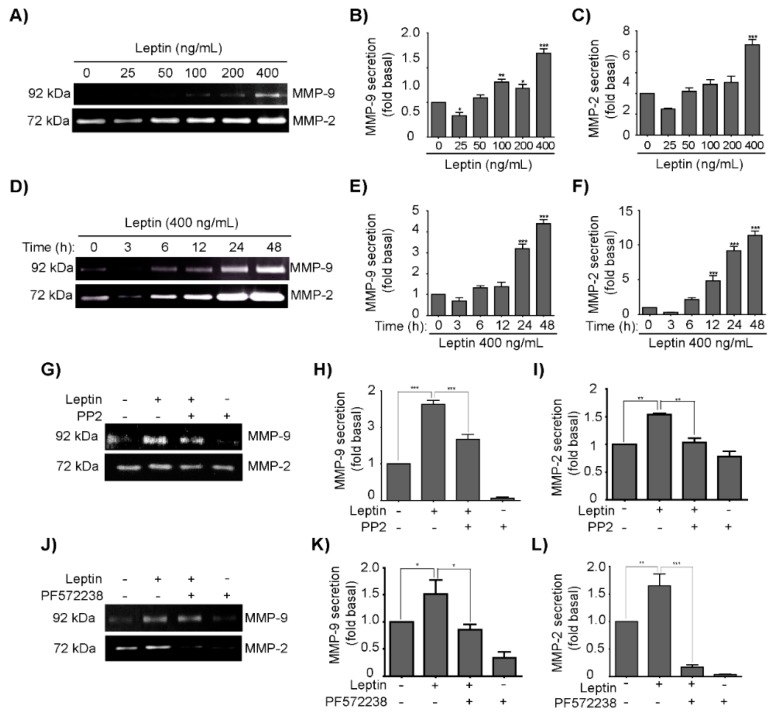
Leptin promotes invasion-related processes by activating MMPs in a Src- and FAK-dependent manner in MCF10A cells. MCF10A cells were treated with leptin 400 ng/mL for different times and MMP activity by degradation of bovine gelatin was evaluated. (**A**) Representative zymogram gels from culture supernatants of MCF10A cells treated with increasing concentrations of leptin corresponding to the degradation bands of MMP-9 (92 kDa) and MMP-2 (72 kDa). The plots represent the densitometric and statistical analyses of the bands obtained by gelatin zymography shown for (**B**) MMP-9, and (**C**) MMP-2. (**D**) Representative zymogram gels from culture supernatants of MCF10A cells treated with 400 ng/mL of leptin for different times corresponding to the degradation bands of MMP-9 and MMP-2. The plots represent the densitometric and statistical analysis of the bands obtained by gelatin zymography shown for (**E**) MMP-9, and (**F**) MMP-2. (**G**) Effect of Src inhibition on MMP-9 and MMP-2 activation. MCF10A cells were treated or not with 400 ng/mL of leptin and 10 µM PP2. The plots represent the densitometric and statistical analysis of the bands obtained by gelatin zymography shown for (**H**) MMP-9, and (**I**) MMP-2 upon inhibition of Src. (**J**) Effect of FAK inhibition on MMP-9 and MMP-2 activation. MCF10A cells were treated or not with 400 ng/mL of leptin and 10 µM PF573228. The plots represent the densitometric and statistical analysis of the bands obtained by gelatin zymography shown for (**K**) MMP-9, and (**L**) MMP-2 upon inhibition of FAK. The values are shown in means ± SD of three independent experiments and are expressed as changes with respect to the control (unstimulated cells). The asterisks indicate the comparison made with respect to the control. * *p* < 0.05, ** *p* < 0.01 and *** *p* < 0.001 by a one-way ANOVA (Dunnett´s and Newman–Keuls’s test).

**Figure 5 cells-08-01133-f005:**
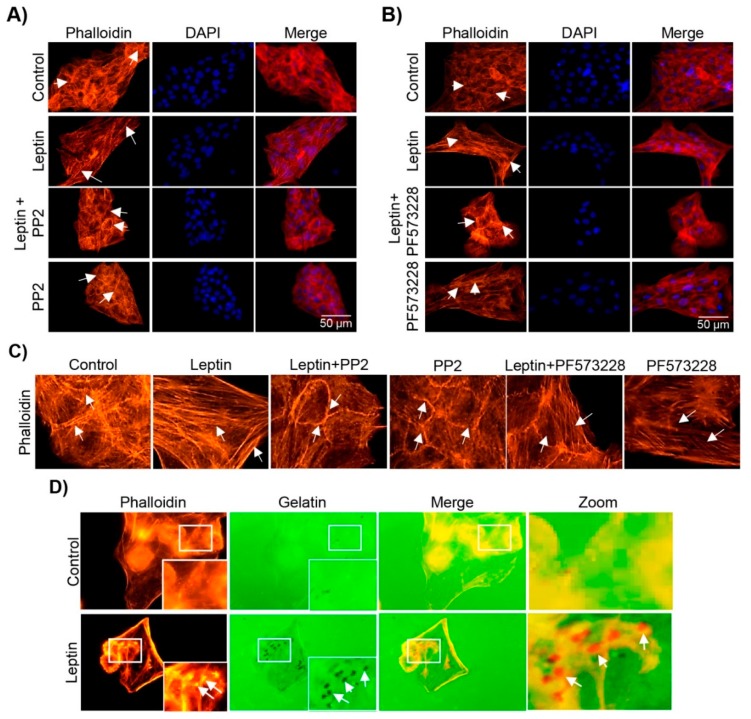
Leptin stimulation leads to the re-arrangement of the actin cytoskeleton and formation of invadopodia. Representative images of MCF10A cells pre-treated or not with PP2 (**A**) or PF-573228 (**B**) for 30 min and subsequently with leptin (400 ng/mL) for 24 h. Actin filaments were detected with Phalloidin-TRITC (red) and DNA was counterstained with DAPI. Images were acquired using the 40× magnification. (**C**) Enlarged images of actin cytoskeleton of MCF10A cells grown in the presence or absence of leptin, and the Src and FAK inhibitors. White arrows indicate structures of actin. (**D**) Representative images of invadopodia formation assays. Actin puncta was detected with phalloidin (red) and Alexa 488-labeled gelatin (green) was used as a specific substrate for the membrane-bound MMP-14 located at the edge of invadopodia. Arrows indicate areas of gelatin degradation. Images were acquired using a 100× magnification.

**Figure 6 cells-08-01133-f006:**
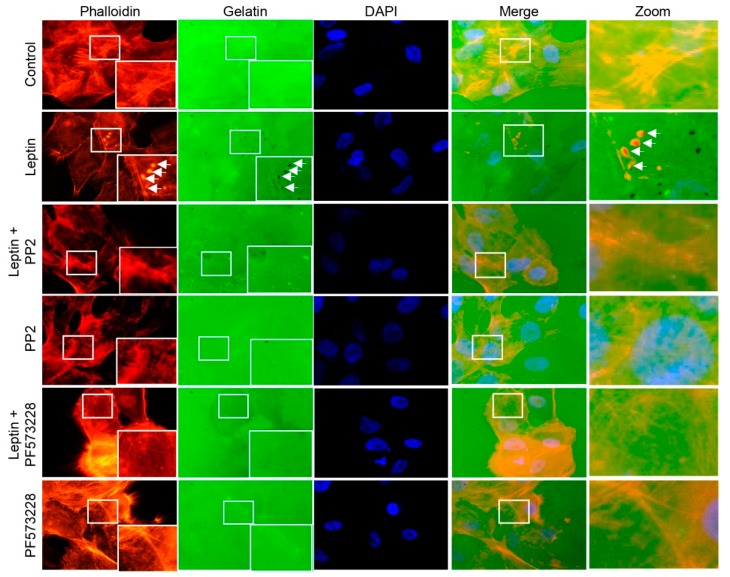
Src and FAK regulate the invasion capacity of MCF10A cells stimulated with leptin. Representative images of MCF10A cells pre-treated or not with PP2 or PF-573228 (10 µM) for 30 min and subsequently incubated in the presence or absence of leptin (400 ng/mL) for 24 h. Actin puncta was detected with phalloidin (red) and Alexa 488-labeled gelatin (green) was used as a specific substrate for the membrane-bound MMP-14 located at the edge of invadopodia. Arrows indicate areas of gelatin degradation. Images were acquired using a 100× magnification.

**Figure 7 cells-08-01133-f007:**
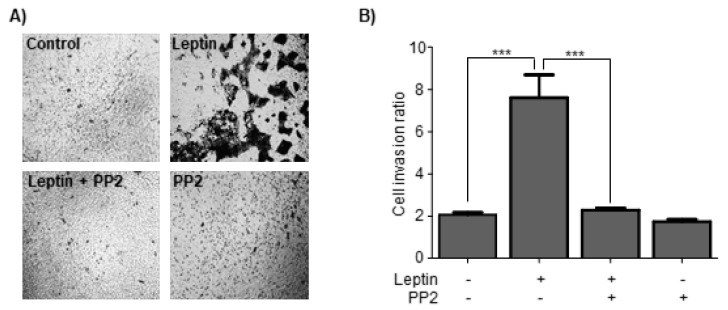
Leptin induce invasion in MCF10A cells in a Src-dependent manner. (**A**) Representative light microscopy images of invasion assays of MCF10A cells pre-treated with AraC and grown in the presence or absence of leptin, and the Src inhibitor PP2. (**B**) Quantitative analyses of the invasion assays presented in (**A**); the values represent the mean ± SD of three independent experiments, and are expressed as a cell invasion ratio (between invading and non-invading cells). The asterisks indicate the comparison made with respect to the control. *** *p* < 0.001 by a one-way ANOVA (Newman-Keuls’s test).

**Figure 8 cells-08-01133-f008:**
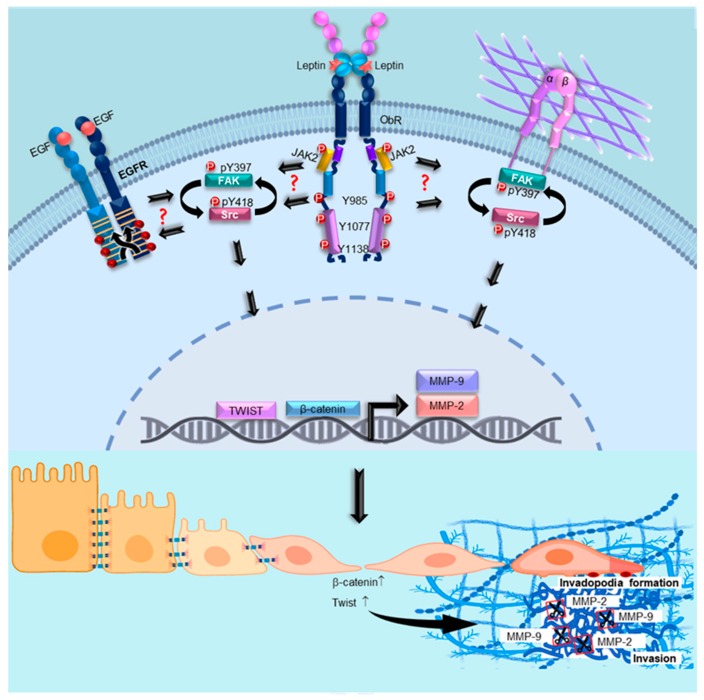
Model for leptin-induced EMT via Src and FAK signaling pathways. Leptin activates FAK and Src in a positive feedback pathway, which in turn regulate the expression and subcellular localization of β-catenin and Twist, promoting the expression of MMP-2 and MMP-9 as well as EMT-related events such as invadopodia formation and cell invasion. ? indicates an unknown mechanism.

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
