# Peer review of "Leptin Promotes Expression of EMT-Related Transcription Factors and Invasion in a Src and FAK-Dependent Pathway in MCF10A Mammary Epithelial Cells"

_cells, 2019, doi:10.3390/cells8101133_

Round 1
Reviewer 1 Report
The present study concerns EMT induced by leptin in normal MCF10A mammary epithelial 4 cells. Authors present their data claiming that leptin acts via FAK and Src kinases influencing the expression of MMPs and select transcription factors (TWIST and beta catenin) leading to invadopodia formation associated with invasion and metastasis. As declared this work extends the authors´ previous paper (Leptin induces partial 609 epithelial-mesenchymal transition in a FAK-ERK dependent pathway in MCF10A mammary 610 non-tumorigenic cells) but, unfortunately, it brings very little novelty. In fact, in the light of the previously published paper, current findings only refine the current ones. In addition, all the work is carried out on one cell line only, which totally insufficient in terms of robustenss of the data - i.e. some more model lines or primary cultures are required to prove authors claims. Moreover, the line MCF10A has been challenged by some authors (PLOS One 2015 Jul 6;10(7):) as being a non-reliable model for human mammary cell studies. Together with a number of minor criticisms - such as non declared used kinase inhibitors concentrations, the absence of verification protocols using siRNa, poor quality of F-actin stained microphotographs (lack of resolution ..., lack of inner scales etc) marks the present work as not acceptable.
Author Response
Response to Reviewer 1:
The present study concerns EMT induced by leptin in normal MCF10A mammary epithelial 4 cells. Authors present their data claiming that leptin acts via FAK and Src kinases influencing the expression of MMPs and select transcription factors (TWIST and beta catenin) leading to invadopodia formation associated with invasion and metastasis. As declared this work extends the authors´ previous paper (Leptin induces partial 609 epithelial-mesenchymal transition in a FAK-ERK dependent pathway in MCF10A mammary 610 non-tumorigenic cells) but, unfortunately, it brings very little novelty. In fact, in the light of the previously published paper, current findings only refine the current ones. In addition, all the work is carried out on one cell line only, which totally insufficient in terms of robustenss of the data - i.e. some more model lines or primary cultures are required to prove authors claims. Moreover, the line MCF10A has been challenged by some authors (PLOS One 2015 Jul 6;10(7):) as being a non-reliable model for human mammary cell studies. Together with a number of minor criticisms - such as non declared used kinase inhibitors concentrations, the absence of verification protocols using siRNa, poor quality of F-actin stained microphotographs (lack of resolution ..., lack of inner scales etc) marks the present work as not acceptable.
Response: We appreciate the opinion of reviewer 1. However, we consider that our research is the first exploration to evaluate the role of FAK and Src kinases in the induction of metastasis-related events and the possible generation of cancer stem cells induced by leptin in a non-tumorigenic cellular model.
We agree with the reviewer that the use of two or more cell lines validates our results. Nevertheless, we consider that the use of this epithelial cell line is appropriate because it is a non-tumor cell line and allowed us to identify the direct effect of leptin on the EMT-related events, without genetic alterations present in mammary cancer cell lines.
We want to point out that the study referred by the reviewer (PLOS One 2015 Jul 6;10(7):) present several differences to our study. In that work, the authors made comparisons in 2D and 3D cultures. However the indicated report did not describe experiments similar to ours, and therefore cannot be deduced to the results obtained in this research.
We appreciate the observations by the reviewer about the quality of the images and methodological points. We want to point out that in “Materials and methods”, specifically in the Cell stimulation section, it was specified the concentrations of both kinases inhibitors. These are further indicated in figure legends and descriptions of results in the revised version of the manuscript. We have also fixed and improved the fluorescent images for F-actin staining for the revised version of our manuscript.

Reviewer 2 Report
Olea-Flores et al seeks to identify mechanisms of leptin mediated invasion and EMT in the MCF710a cell line. The results here are compelling but require clarification on exact mechanism of action due to the possibility of off target effects from inhibitors used.
Major comments:
PP2 is demonstrated to inhibit cell proliferation in other cell lines, authors should demonstrate that treatment of PP2 does not result in loss of cell number to solidify that the observed effect is through invasion and not differences in proliferation. PP2 has targets other than Src, authors should disclose concertation used for this drug and either identify how they accounted for/ruled out off target kinases as means of action or include possibility of effects in discussion. Figure 1 suggests a change in total SRC at 60 minutes. Can authors explain this? Quantification for total protein changes should be included to further validate phospho changes observed.
Minor comments
The GAPDH loading control is variable across leptin stimulation. Is this a representation of the effects of leptin (as leptin is described regulator of GAPDH in other species) or is this there a better representative image for protein loading? Authors should include a better representation of western blot without dark spots (Figure 2) if possible.Author Response
Response to Reviewer 2:
Olea-Flores et al seeks to identify mechanisms of leptin mediated invasion and EMT in the MCF710a cell line. The results here are compelling but require clarification on exact mechanism of action due to the possibility of off target effects from inhibitors used.
Major comments:
PP2 is demonstrated to inhibit cell proliferation in other cell lines, authors should demonstrate that treatment of PP2 does not result in loss of cell number to solidify that the observed effect is through invasion and not differences in proliferation.
Response: We thank the reviewer for this observation and apologize for the lack of clarity in the description of the methods and results. In the invasion protocol the AraC proliferation inhibitor is used for all the conditions tested. Therefore, we can conclude that the observed effects are due to the treatment of PP2. We have added text in these two sections to make this point clear.
PP2 has targets other than Src, authors should disclose concertation used for this drug and either identify how they accounted for/ruled out off target kinases as means of action or include possibility of effects in discussion.
Response: We apologize again for the lack of clarity. The concentration used was 10 uM. We have now added this information in all the pertinent sections of the revised version of the manuscript. We have also indicated in the discussion section that additional downstream effectors may be involved in this process.
Figure 1 suggests a change in total SRC at 60 minutes. Can authors explain this? Quantification for total protein changes should be included to further validate phospho changes observed.
Response: We apologize for the poor quality of the initial western blots presented in the first version of our manuscript. We have now performed additional WBs and included better images. We hope these new results satisfy the reviewers’ concerns.
The GAPDH loading control is variable across leptin stimulation. Is this a representation of the effects of leptin (as leptin is described regulator of GAPDH in other species) or is this there a better representative image for protein loading? Authors should include a better representation of western blot without dark spots (Figure 2) if possible.
Response: Again, we thank the reviewer for pointing this out, and apologize for the poor quality of the western blots presented in the initial version of our manuscript. We have now performed additional experiments and included better images. We hope these new results satisfy the reviewers’ concerns.

Reviewer 3 Report
This manuscript builds on previous research from the same authors (ref 43), showing that leptin induces epithelial to mesenchymal transition (EMT) in MCF10A. In Ref 43, leptin induced FAK and Erk phosphorylation, elevated vimentin expression and cytoplasmic E-cadherin localization. In addition, leptin-induced cell migration was blocked with FAK and Erk inhibitors. In this manuscript the authors provide additional changes that occur in MCF10A cells upon leptin exposure. These observations are rather descriptive and provide additional support for leptin-induced and FAK and Src-dependent EMT. However, Leptin-induced Src signalling has also been observed by others (e.g. Plos One 2017 Mar 1;12(3):e0170675; Breast Cancer Res 2014 Oct 25;16(5):426). How leptin activates signalling cascades that drive EMT remains unclear. In fact, even the transactivation of RTKs via elevated leptin levels cannot be excluded (Curr Pharm Des. 2014;20(4):616-24). Some comments that should be addressed are listed below:
Fig. 1 shows leptin-induced activation of Src (A, C) and FAK (B, D). Fig. 1B and 1D is similar to data previously provided in Ref 43. However, the magnitude and kinetics of FAK activation in Fig. 1B and 1D is different to Ref 43. This should be clarified. Leptin-induced Src activation peaks at 15 min, indicating that leptin-induced FAK activation (peak at 10 min) might occur upstream of Src phosphorylation. As Src and FAK inhibitors (PP2, PF573228) are used throughout the manuscript, clarifying pY-397 FAK levels in the presence of Src inhibitor and pY416-Src levels in PF573228 incubated cells would contribute to the mechanism. Fig. 2A-C supports leptin-inducible changes in Twist and b-catenin expression and localization observed by others (ref 9, 10). Why are the kinetics of leptin-inducible Twist and b-catenin expression different? Ref 9 identifies leptin to stimulate b-catenin via GSK3β and Akt, while Ref 10 describes PMK2 as critical for leptin-induced changes in EMT marker expression. Can these signalling cascades be excluded in the data shown here? Fig. 2D-E shows Twist and b-catenin localization in leptin-incubated cells. In lane 242, it is stated that Twist is largely located in the nucleus at all times. For this reviewer, this was not apparent for images at t = 3 and 6 hours. Quantification and arrows should be considered to clarify the location of Twist over time. Likewise, arrows/arrowheads to highlight nuclear and membrane location of b-catenin would be helpful. If b-catenin expression increases 3-fold after 48 h (2C), why is this not reflected in the immunostaining intensity shown in Fig. 2E? Fig. 3 shows leptin-inducible Twist and b-catenin expression with and without Src and FAK inhibitor. Control blots showing loss of pY416-Src and pY397-FAK in the presence of inhibitors should be provided. An improved/clearer WB in Fig. 3A, lane 3 should be provided. If Twist is still expressed under these conditions, why is there no Twist immunostaining (Fig. 3G) under these conditions? The same applies for the mismatch of WB signals/intensity profiles for PF573228-incubated cells with those after immunostaining in 3G, which should be comparable to the control. Fig. 4I and 4L: The Y-axis is labelled ‘MMPs’? Should this be MMP2 or MMP9? Please clarify here and in the figure legend. Fig. 5: For this reviewer, even after magnification of images in Panel A and B, ‘more and thicker actin filaments with a linear distribution etc (lane 340-342) were not clearly visible. Likewise, ‘Src had an effect on the formation of thick fibers’ (lane 372). These images should be improved and areas of interest should be highlighted or enlarged. Fig. 8: A more detailed figure legend providing an explanation for the various arrows and symbols should be added.Minor:
lane 93: font size of references 36-39 lane 113: rabbit? Lane 340: as expected the leptin treated cells treated with leptin?? lane 450…where it actsAuthor Response
Response to Reviewer 3:
This manuscript builds on previous research from the same authors (ref 43), showing that leptin induces epithelial to mesenchymal transition (EMT) in MCF10A. In Ref 43, leptin induced FAK and Erk phosphorylation, elevated vimentin expression and cytoplasmic E-cadherin localization. In addition, leptin-induced cell migration was blocked with FAK and Erk inhibitors. In this manuscript the authors provide additional changes that occur in MCF10A cells upon leptin exposure. These observations are rather descriptive and provide additional support for leptin-induced and FAK and Src-dependent EMT. However, Leptin-induced Src signalling has also been observed by others (e.g. Plos One 2017 Mar 1;12(3):e0170675; Breast Cancer Res 2014 Oct 25;16(5):426). How leptin activates signalling cascades that drive EMT remains unclear. In fact, even the transactivation of RTKs via elevated leptin levels cannot be excluded (Curr Pharm Des. 2014;20(4):616-24). Some comments that should be addressed are listed below:
Fig. 1 shows leptin-induced activation of Src (A, C) and FAK (B, D). Fig. 1B and 1D is similar to data previously provided in Ref 43. However, the magnitude and kinetics of FAK activation in Fig. 1B and 1D is different to Ref 43. This should be clarified
Response: We apologize for the confusion. In the indicated manuscript, the amount of protein loaded was lower (10 ug) than in the current work (20 ug). We decided to load more protein for this new study to show a better representation of the results obtained with FAK. As the reviewer noted, the levels of FAK are barely distinguishable in our previous work. We have included the amount of protein used for these new experiments in the methods section of the present manuscript.
Leptin-induced Src activation peaks at 15 min, indicating that leptin-induced FAK activation (peak at 10 min) might occur upstream of Src phosphorylation. As Src and FAK inhibitors (PP2, PF573228) are used throughout the manuscript, clarifying pY-397 FAK levels in the presence of Src inhibitor and pY416-Src levels in PF573228 incubated cells would contribute to the mechanism.
Response: We thank the reviewer for this suggestion. We have added text to clarify the role of PP2 and PF573228 in the FAK and Src kinases activation. In addition we performed new reciprocal western blots for the kinases, in the presence of the inhibitors (new figure 3). These new findings allow us to report that FAK and Src activation respond to a positive feedback between both kinases. We hope these new data satisfy the reviewer’s concerns
Fig. 2A-C supports leptin-inducible changes in Twist and b-catenin expression and localization observed by others (ref 9, 10). Why are the kinetics of leptin-inducible Twist and b-catenin expression different?
Response: The indicated studies used breast cancer cell lines. These cells have an altered genetic background which could be related to the differences in Twist and b-catenin expression and localization obtained in the MCF10A cell line, which is a non-tumorigenic cell line. We believe that these differences observed are dependent on the cellular context of the different models used. Text has been added in the discussion to clarify this point.
Ref 9 identifies leptin to stimulate b-catenin via GSK3β and Akt, while Ref 10 describes PMK2 as critical for leptin-induced changes in EMT marker expression. Can these signalling cascades be excluded in the data shown here?
Response: We definitely cannot exclude the possibility that GSK3b, Akt or PKM2 are involved in this process. The various mechanisms for gene expression may involve the activation of many kinases that regulate the expression and activity of proteins and biological events during EMT. Our group is currently working to elucidate additional signaling pathways as those indicated by the reviewer, and we believe these are beyond the scope of this manuscript. However, we added text to the discussion to highlight the point made by the reviewer.
Fig. 2D-E shows Twist and b-catenin localization in leptin-incubated cells. In lane 242, it is stated that Twist is largely located in the nucleus at all times. For this reviewer, this was not apparent for images at t = 3 and 6 hours. Quantification and arrows should be considered to clarify the location of Twist over time. Likewise, arrows/arrowheads to highlight nuclear and membrane location of b-catenin would be helpful.
Response: We thanks the reviewer for pointing this out. Arrows have been added to the figures and text has been modified accordingly in the revised version of the manuscript.
If b-catenin expression increases 3-fold after 48 h (2C), why is this not reflected in the immunostaining intensity shown in Fig. 2E?
Response: In our opinion, the increase in b-catenin is evident at 48 h of stimulation with leptin and its cellular localization is mostly in the nucleus. The fluorescent images were acquired using the same parameters in the microscope, and figure 2 panel E reflects that increase. We apologize for the poor quality of the figures presented initialy.
Fig. 3 shows leptin-inducible Twist and b-catenin expression with and without Src and FAK inhibitor. Control blots showing loss of pY416-Src and pY397-FAK in the presence of inhibitors should be provided. An improved/clearer WB in Fig. 3A, lane 3 should be provided. If Twist is still expressed under these conditions, why is there no Twist immunostaining (Fig. 3G) under these conditions? The same applies for the mismatch of WB signals/intensity profiles for PF573228-incubated cells with those after immunostaining in 3G, which should be comparable to the control.
Response: We thank the reviewer for this observation and apologize again for the poor quality of the figures chosen. We have included better images and the requested images such as Control blots showing loss of pY416-Src and pY397-FAK in the presence of inhibitors, PP2 and PF573228. In our opinion, the levels of Twist where inhibitors were used are proportional between the WB and IF assays. We hope that all the new images satisfy the reviewer’s concerns.
Fig. 4I and 4L: The Y-axis is labelled ‘MMPs’? Should this be MMP2 or MMP9? Please clarify here and in the figure legend.
Response: We apologize for this mistake, and thank the reviewer for noticing. We have fixed the Y-axis and corresponds to MM-2 and MMP-9.
Fig. 5: For this reviewer, even after magnification of images in Panel A and B, ‘more and thicker actin filaments with a linear distribution etc (lane 340-342) were not clearly visible. Likewise, ‘Src had an effect on the formation of thick fibers’ (lane 372). These images should be improved and areas of interest should be highlighted or enlarged.
Response: again we thank the reviewer for this suggestion and apologize for the poor quality of images chosen. The images were improved and the areas of interest were enlarged.
Fig. 8: A more detailed figure legend providing an explanation for the various arrows and symbols should be added.
Response: The reviewer’s observation was considered and the figure legend was modified accordingly.
Minor:
lane 93: font size of references 36-39 lane 113: rabbit? Lane 340: as expected the leptin treated cells treated with leptin?? lane 450…where it acts.
Response: All minor revisions were fixed. Thank you!

Round 2
Reviewer 2 Report
Reviewer comments have been sufficiently met.
Author Response
We thank the reviewer for his/her valuable opinion on our work and are grateful for the positive comments made by the reviewer.

Reviewer 3 Report
This is a greatly improved manuscript, and all concerns were addressed. Although the novelty of this study remains a bit limited, the experiments are now clearer and better controls are provided. This reviewer has no further concerns.
Author Response
We appreciate the suggestions and comments of the reviewer since they allowed us to improve this manuscript. Thank you.
